# Exploring the Relationship of rs2802292 with Diabetes and NAFLD in a Southern Italian Cohort—Nutrihep Study

**DOI:** 10.3390/ijms25179512

**Published:** 2024-09-01

**Authors:** Giovanna Forte, Rossella Donghia, Martina Lepore Signorile, Rossella Tatoli, Caterina Bonfiglio, Francesco Losito, Katia De Marco, Andrea Manghisi, Filomena Anna Guglielmi, Vittoria Disciglio, Candida Fasano, Paola Sanese, Filomena Cariola, Antonia Lucia Buonadonna, Valentina Grossi, Gianluigi Giannelli, Cristiano Simone

**Affiliations:** 1Medical Genetics, National Institute of Gastroenterology, IRCCS “Saverio de Bellis” Research Hospital, 70013 Castellana Grotte, Italy; giovanna.forte@irccsdebellis.it (G.F.); martina.lepore@irccsdebellis.it (M.L.S.); katia.demarco@irccsdebellis.it (K.D.M.); andrea.manghisi@irccsdebellis.it (A.M.); floranna.guglielmi@irccsdebellis.it (F.A.G.); vittoria.disciglio@irccsdebellis.it (V.D.); candida.fasano@irccsdebellis.it (C.F.); paola.sanese@irccsdebellis.it (P.S.); filo.cariola@irccsdebellis.it (F.C.); lucia.buonadonna@irccsdebellis.it (A.L.B.); 2Data Science Unit, National Institute of Gastroenterology, IRCCS “Saverio de Bellis” Research Hospital, 70013 Castellana Grotte, Italy; rossella.donghia@irccsdebellis.it (R.D.); rossella.tatoli@irccsdebellis.it (R.T.); catia.bonfiglio@irccsdebellis.it (C.B.); 3Gastroenterology Unit, National Institute of Gastroenterology, IRCCS “Saverio de Bellis” Research Hospital, 70013 Castellana Grotte, Italy; francesco.losito@irccsdebellis.it; 4Scientific Direction, National Institute of Gastroenterology, IRCCS “Saverio de Bellis” Research Hospital, 70013 Castellana Grotte, Italy; gianluigi.giannelli@irccsdebellis.it; 5Medical Genetics, Department of Precision and Regenerative Medicine and Jonic Area (DiMePRe-J), University of Bari Aldo Moro, 70124 Bari, Italy

**Keywords:** single nucleotide polymorphisms (SNPs), type 2 diabetes, non-alcoholic fatty liver disease (NAFLD)

## Abstract

**Background:** The minor G-allele of FOXO3 rs2802292 is associated with human longevity. The aim of this study was to test the protective effect of the variant against the association with type 2 Diabetes and NAFLD. **Methods:** rs2802292 was genotyped in a large population of middle-aged subjects (n = 650) from a small city in Southern Italy. All participants were interviewed to collect information about lifestyle and dietary habits; clinical characteristics were recorded, and blood samples were collected from all subjects. The association between rs2802292 and NAFLD or diabetes was tested using a logistic model and mediation analysis adjusted for covariates. **Results:** Overall, the results indicated a statistical association between diabetes and rs2802292, especially for the TT genotype (OR = 2.14, 1.01 to 4.53 95% C.I., *p* = 0.05) or in any case for those who possess the G-allele (OR = 0.45, 0.25 to 0.81 95% C.I., *p* = 0.008). Furthermore, we found a mediation effect of rs2802292 on diabetes (as mediator) and NAFLD. There is no direct relationship between rs2802292 and NAFLD, but the effect is direct (β = 0.10, −0.003 to 0.12 95% C.I., *p* = 0.04) on diabetes, but only in TT genotypes. **Conclusions:** The data on our cohort indicate that the longevity-associated FOXO3 variant may have protective effects against diabetes and NAFLD.

## 1. Introduction

Approximately one-third of the variability in the human lifespan is affected by genetic factors [1,2]. It is already known that the lifespan is influenced by insulin resistance, which is associated with diabetes and cardiovascular diseases that significantly influence mortality and, hence, longevity [3]. Forkhead box O3 (FOXO3) is a transcription factor, evolutionarily conserved from Caenorhabditis elegans to mammals, involved in the insulin signaling pathway [4].

It has been reported that FOXO3 regulates the expression of genes controlling many processes, including cell cycle gene expression [5], DNA damage repair [6], energy metabolism [7], oxidative stress resistance [8], and apoptosis [9] that promote health and a long lifespan [10,11].

Several studies have found an association of single nucleotide polymorphisms (SNPs) rs2802292 of FOXO3 with human longevity; this sequence lies in the intronic region (intron 2) of FOXO3 [10,12]. The G-allele of rs2802292 is associated with a considerable reduction of mortality risk and different major age-associated clinical causes of death, such as coronary heart disease, stroke, and cancer [13].

Recently, we found a remarkably higher cancer risk for Peutz-Jeghers and the PTEN hamartoma tumor syndromes in TT genotype carriers of rs2802292 compared to patients carrying at least one G-allele [14]. Moreover, we performed an extensive molecular characterization of human intestinal cell lines, showing that the G-allele was correlated with increased mRNA and protein expression of FOXO3 [14]. Importantly, several other studies confirmed that the intronic rs2802292 G-allele was correlated with an enhanced expression of FOXO3, suggesting that intron 2 of FOXO3 may be a regulatory region [13,15,16,17]. Lately, we found that a specific sequence of 90-bp around rs2802292 has enhancer functions and that the G-allele generates a unique heat shock element (HSE) binding site for HSF1, which induces FOXO3 expression in response to various stress stimuli [15]. Specifically, HSF1 promotes the interaction of the 5UTR promoter region with the rs2802292 enhancer region at the FOXO3 locus. The presence of this HSF1-FOXO3 axis is involved in different stress response pathways, promoting reactive oxygen species (ROS) detoxification, redox balance, and DNA repair, leading to an increased lifetime [15].

Importantly, the rs2802292 G-allele of FOXO3 results are associated with improved peripheral and hepatic insulin sensitivity. Indeed, the enhanced gene transcription mediated by FOXO3 overexpression in G-allele carriers showed beneficial effects on glucose metabolism [16].

An imbalance in glucose metabolism leads to metabolic disorders like diabetes and obesity [17]. Currently, about 60 million people live with diabetes in Europe, and its prevalence is increasing among all ages, mostly due to a general increase in overweight and obesity, unhealthy dietary habits, and physical inactivity. As a consequence, diabetes deaths will likely double by 2030 compared to 2005 [18].

In addition, growing epidemiological evidence suggests a two-way relationship between diabetes and non-alcoholic fatty liver disease (NAFLD). On one hand, diabetes is considered a risk factor for the development of NAFLD and the progression to more advanced liver disease, including fibrosis, cirrhosis, and hepatocellular carcinoma [19,20,21,22,23]. On the other hand, NAFLD may precede and/or promote the development of type 2 diabetes [24].

The aim of this study was to investigate the relationship between FOXO3 SNP rs2802292 and diabetes and NAFLD in a previously well-studied large population of middle-aged subjects. To this end, we genotyped this study population to assess epidemiological and physiopathological parameters.

## 2. Results

The epidemiological and clinical characteristics of the NUTRIHEP cohort at the second recall for the total cohort and stratified sub-cohort for genotypes—homozygous minor (GG), heterozygous (GT), and homozygous major (TT)—are reported in Table 1.

Males were less prevalent than females (42.15%) in the total cohort, and the mean age was 61.25 ± 11.91 years. Secondary level of education was predominant (32.90%), and 81.94% of subjects were married or cohabiting. Only 11.40% have a smoking habit. All participants were alive on 31 December, 2023, and it can be considered a working-age population.

The prevalence of diabetes changed across genotype categories and is statistically significant (*p* = 0.02), with a higher prevalence in subjects with the TT genotype compared to heterozygotes (14.38% vs. 6.89%, *p* = 0.005). Regarding blood parameters, there were statistically significant differences between groups for cholesterol (*p* = 0.03), with lower values in TT but also in comparisons between patients who are homozygous for GG and heterozygous GT (187.69 ± 37.79 vs. 197.93 ± 38.22 and 187.69 ± 37.79 vs. 196.86 ± 34.40, *p* = 0.02 to *p* = 0.03 respectively). Monocytes % outside the range show an opposite trend, where the highest prevalence is found in TT (4.58%. *p* = 0.009). On the contrary, SGPT in the same group had a prevalence of 6.54% (*p* = 0.03), with a statistically significant difference between TT and GT (6.54% vs. 14.67%, *p* = 0.01). As to the parameters linked to liver function, ceruloplasmin has fairly comparable concentrations between TT and GT but a statistically significant variation between the three groups (*p* = 0.04), showing lower values from GT to GG (30.49 ± 6.43 vs. 31.83± 6.24, *p* = 0.02). The same significance was observed in the same variable used as a categorical variable, both as regards the prevalences between groups (*p* = 0.01) and between GT and GG (20.36% vs. 9.88%, *p* = 0.004). Regarding the concentration of α1AT, heterozygotes appear to have lower concentrations than their homozygous counterparts (GG) (171.00 ± 27.82 vs. 175.10 ± 28.26, *p* = 0.04).

The Hardy-Weinger equilibrium (HWE) was not significant (*p* = 0.53), indicating that there is no significant difference between observed and expected genotype frequencies, suggesting that this cohort is in Hardy-Weinberg equilibrium and the assumptions of HWE are met.

The association between diabetes and rs2802292 in the model adjusted for age and gender is shown in Table 2.

A strong association between diabetes and rs2802292 was found (OR = 2.14, 1.01 to 4.53 95% C.I., *p* = 0.05), underlining that the TT genotype is a risk factor for developing diabetes. To highlight the role of G-alleles, a variable of G-carriers was built. It was found that subjects with the GG or GT genotypes seem to be protected against diabetes (OR = 0.45, from 0.25 to 0, 81 95% CI, *p* = 0.008).

In the mediation analysis (Table 3), we found a mediation effect of rs2802292 on diabetes (as mediator) and NAFLD.

There is no direct relationship of rs2802292 with NAFLD, but the effect is direct (β = 0.10, −0.003 to 0.12 95% C.I., *p* = 0.04) on diabetes but only in TT genotypes. A direct relationship was found between diabetes and NAFLD (β = 0.13, 0.10 to 0.35 95% C.I., *p* < 0.001) (Figure 1).

To summarize and visualize the information about categorical blood parameters stratified for genotypes, Multiple Correspondence Analysis (MCA) was used to visualize subject profiles (Appendix A).

## 3. Discussion

In invertebrates, the FOXO3 homolog DAF-16 has been found to increase lifespan and regulate insulin pathways. In mammals, FOXO3 plays a role in stress resistance, cell proliferation/arrest, survival/death, metabolism, and autophagy; moreover, FOXO3 is involved in tumor suppression, regulation of energy metabolism, development of specific tissues, genomic integrity preservation, and a reduction of oncogenic mutation accumulation [6,25,26,27,28,29,30,31]. FOXO3 also protects cells, promoting a response to ROS accumulation through gene regulation. Indeed, the FOXO3 upregulation in response to ROS accumulation leads to an increased expression of SOD2 and catalase, which is involved in cell detoxification and cell survival [8]. The rs2802292 G-allele at the FOXO3 locus located in intronic regions of the gene has been shown to be associated with longevity in different human populations [10,32,33,34,35]. The SNP correlates with reduced age-related diseases in centenarians [14,15]. Notably, the longevity-related function of DAF-16 in C. elegans is evolutionarily conserved [36]. The longevity-associated genetic variants in FOXO3 are associated with a mortality risk reduction for coronary heart disease [13,37,38] and with blood pressure and essential hypertension reduction in American women [39,40,41]. Moreover, rs2802292 FOXO3 was demonstrated to have enhancer functions, creating a binding site for HSF1 in response to stress stimuli and inducing survival by promoting ROS detoxification, redox balance, and DNA repair [15]. Individuals carrying the minor G-allele of FOXO3 rs2802292 showed improved peripheral and hepatic insulin sensitivity, suggesting that the effect on glucose metabolism could be mediated by enhanced gene transcription. In fact, the rs2802292 G-allele has been associated with a decreased risk of fasting hyperglycemia, lower levels of clinical biomarkers including fasting glucose and hemoglobin A1c, improved insulin resistance (HOMA-IR) only in long-lived individuals [42]. Moreover, the rs2802292 G-allele of FOXO3 is associated with lower blood glucose levels in older women with type 2 diabetes in East China [43]. In this study, for the first time, we observed a relationship between FOXO3 SNP rs2802292 and diabetes and NAFLD in a large population of middle-aged subjects. The variant is associated with diabetes, showing a higher prevalence in subjects with the TT genotype compared to heterozygotes (14.38% vs. 6.89%, *p* = 0.005). Profiling of plasma biomarkers revealed that the percentage of monocytes outside the range has the highest prevalence in TT (4.58%. *p* = 0.009). Indeed, higher levels of monocytes have been implicated in the pathology of numerous inflammatory and autoimmune diseases [44]. Recently, a heightened monocyte proinflammatory/cytolytic activity has been associated with Type 1 Diabetes susceptibility and progression [45]. Regarding the parameters linked to liver function, ceruloplasmin has an important role in iron metabolism and oxidase activity. Serum ceruloplasmin can remove ROS, and its anti-inflammatory activity results in increased levels of an acute response protein in the context of inflammatory diseases [46]. Our findings revealed significantly lower values of ceruloplasmin between GT and GG, which could ameliorate the liver function.

Our study found a strong association between diabetes and the non-protective FOXO3 genotype (TT). Subjects with GG or GT genotypes are protected from diabetes. Interestingly, the mediation plot between rs2802292, diabetes, and NAFLD showed a direct effect with NAFLD only in TT genotypes.

In conclusion, the present study extends our findings of an association of protective effects of the G-allele of FOXO3 against diabetes and NAFLD, highlighting its relative importance compared with the well-established risk factors. It also suggests the possibility that FOXO3 may be a potential target for intervention to protect subjects at increased risk (TT genotype) in which a specific dietary regimen and physical activity programs could prevent diabetes and reduce the steatosis risk.

## 4. Materials and Methods

### 4.1. Study Population

The NUTRIHEP study cohort was first extracted in 2005–2006 from the medical records of general practitioners in the municipality of Putignano (>18 years) (Bari), a small city in southern Italy about 20 km from the coast. Participants were first interviewed (n = 2550) in 2004–2005 by trained physicians to collect information on their sociodemographic, clinical characteristics and dietary habits [47]. From 2014 to 2018, all eligible subjects, starting with NUTRIHEP participants, were invited to participate in the follow-up, and adherence was 86.08% (n = 2195). From these patients, 650 samples were stored for future molecular analysis and are included in this study (Figure 2).

All participants signed informed consent after receiving complete information on the medical data to be studied. The study was approved by the Ethics Committee of the Ministry of Health (DDG-CE-502/2005; DDG-CE-792/2014, 20 May 2005 and 14 February 2014 respectively). Overall mortality was updated to 31 December 2023.

### 4.2. Lifestyle, Clinical, and Dietary Assessment

Lifestyle and anthropometric assessments were performed by clinicians during examination at the study center. Smoking status was based on the single question, “Do you smoke?”. The level of education was expressed as years of schooling. Weight was measured with an electronic scale (SECA©) and recorded to the nearest 0.1 kg. Height was measured with a wall-mounted stadium meter (SECA©) and recorded to the nearest 1 cm. Body mass index (BMI) was calculated as kg/m^2^. Height and weight measurements were performed using a SECA© stadium meter. Blood was collected from all subjects in the morning after an overnight fast. Aliquots were stored in the biobank according to validated protocols and processed by expert personnel. For all blood parameters, the categorical type (out of range) was built based on the normal ranges at the laboratory that conducted the assays. Diabetes has been diagnosed based on drug treatment or endocrinologist referral attesting to the pathology, while NAFLD had been diagnosed in subjects with hepatic steatosis on ultrasound [48].

### 4.3. DNA Extraction and TaqMan PCR Assay

Genomic DNA was extracted from peripheral blood with the MagCore^®^ Genomic DNA Whole Blood Kit (MGB400-08, Amerigo Scientific) according to the manufacturer’s instructions. Genotyping FOXO3 rs2802292 was performed using TaqMan SNP Genotyping Assays (ID: C__16097219_10, Applied Biosystems). Briefly, 5 ng of DNA was amplified for each sample in a total volume of 10 μL using TaqMan™ Genotyping Master Mix (4371353, Applied Biosystems) according to the manufacturer’s instructions. The primer and probe sequences and PCR conditions are available upon request (https://www.thermofisher.com/pl/en/home.html, accessed on 1 June 2024). Randomly selected samples were verified using Sanger sequencing.

### 4.4. Statistical Analysis

Patient characteristics are reported as mean and standard deviation (M ± SD) for continuous variables and as frequency and percentages (%) for categorical variables. To test the association between genotypes group (GG, GT, and TT), the Chi-square test was used for categorical variables, while the Kruskal–Wallis equality rank test was used to compare more than two independent groups. Dunn’s test and the proportion test were performed for multiple pairwise comparisons, with Bonferroni adjustment.

The distribution of allelic frequencies in the total cohort was studied, and deviations from Hardy-Weinger equilibrium (HWE) were evaluated. To evaluate the association of diabetes on rs2802292 (also built as a combination), a logistic regression model, adjusted for age and gender, was built. The estimators were reported as odds ratios (ORs) and 95% confidence intervals. Generalized Causal Mediation Analysis Mediation analysis [49] was made to assess the direct and indirect effects of NAFLD on rs2802292 and diabetes as a mediator, adjusted for age and gender.

To summarize and visualize the information about blood parameters stratified by genotypes and to create a profile of patients, Multiple Correspondence Analysis (MCA) was performed. To test the null hypothesis of no association, the two-tailed probability level was set at <0.05. The analyses were conducted with StataCorp. 2023. Stata Statistical Software: Release 18. College Station, TX, USA: StataCorp LLC., and jamovi (version 2.3.28), while RStudio (“Chocolate Cosmos” Release) was used for the plots.

## Figures and Tables

**Figure 1 ijms-25-09512-f001:**
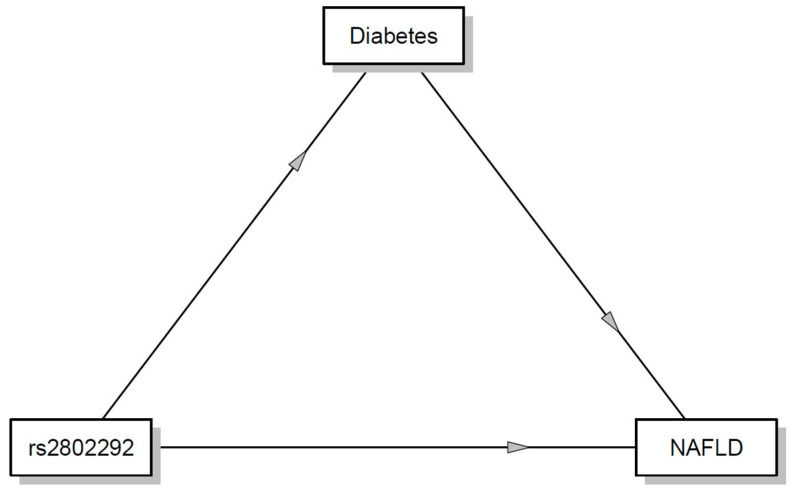
Mediation plot between rs2802292, diabetes, and NAFLD.

**Figure 2 ijms-25-09512-f002:**
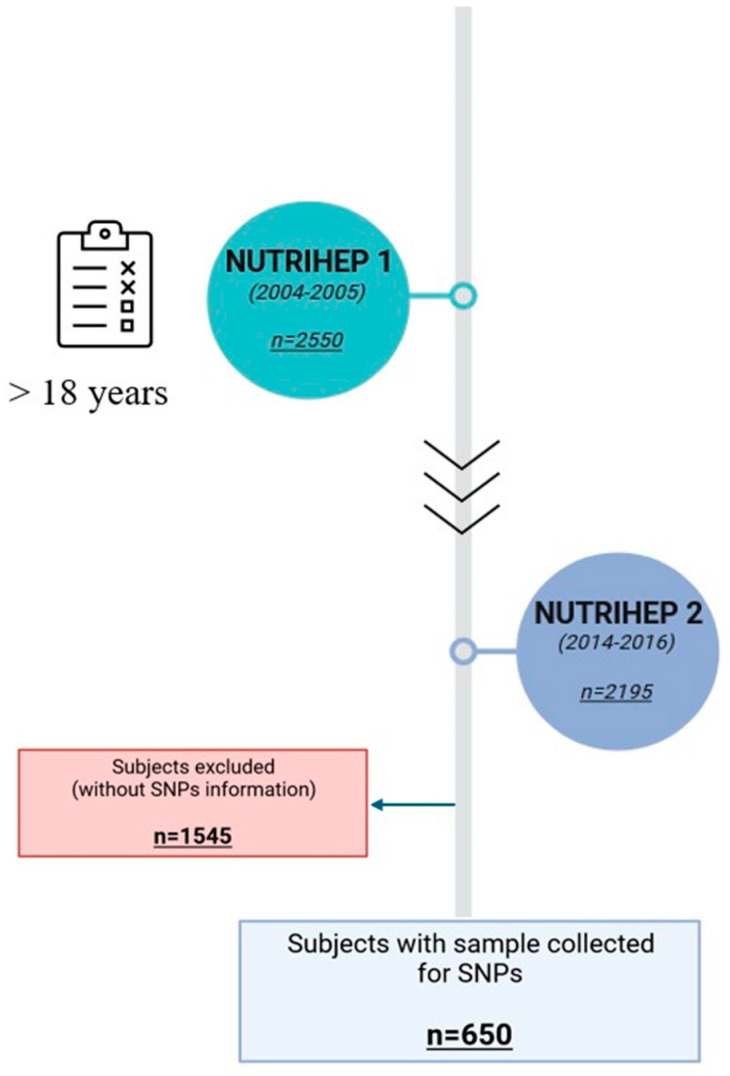
Flowchart of NUTRIHEP study. Image created with BioRender (accessed on 14 June 2024).

**Table 1 ijms-25-09512-t001:** Epidemiological and clinical parameters comparison between different genotypes in the total cohort (n = 650).

Parameters *	Total Cohort (n = 650)	Genotypes			p ^^^	Multiple Comparisons ^Ψ^		
		Homozygous Minor (GG) (n = 163) (a)	Heterozygous (GT) (n = 334) (b)	Homozygous Major (TT) (n = 153) (c)		(b) vs. (a)	(c) vs. (a)	(c) vs. (b)
Gender (M) (%)	274 (42.15)	65 (39.88)	142 (42.51)	67 (43.79)	0.77 ^†^	0.99 ^¥^	0.99 ^¥^	0.99 ^¥^
Age (yrs)	61.25 ± 11.91	61.93 ± 12.50	60.70 ± 11.73	61.72 ± 11.68	0.30	0.46	0.99	0.23
Age Classes (%) (yrs)					0.37 ^†^	0.99 ^¥^	0.99 ^¥^	0.50 ^¥^
Adults (≤65)	394 (60.62)	98 (60.12)	210 (62.87)	86 (56.21)				
Elderly (>65)	256 (39.38)	65 (39.88)	124 (37.13)	67 (43.79)				
Education (%)					0.08 ^†^	0.99 ^¥^	0.99 ^¥^	0.99 ^¥^
None	14 (2.26)	4 (2.60)	4 (1.25)	6 (4.08)				
Elementary School	193 (31.13)	44 (28.57)	99 (31.03)	50 (34.01)				
Secondary School	204 (32.90)	51 (33.12)	101 (31.66)	52 (35.37)				
High School	163 (26.29)	43 (27.92)	90 (28.21)	30 (20.41)				
Degree	22 (3.55)	7 (4.55)	15 (4.70)	0 (0.00)				
Post-Degree	24 (3.87)	5 (3.25)	10 (3.13)	9 (6.12)				
Civil Status (%)					0.68 ^†^	0.99 ^¥^	0.89 ^¥^	0.57 ^¥^
Single	48 (7.74)	14 (9.09)	26 (8.15)	8 (5.44)				
Married or Cohabiting	508 (81.94)	126 (81.82)	257 (80.56)	125 (85.03)				
Divorced or Separated	17 (2.74)	3 (1.95)	12 (3.76)	2 (1.36)				
Widow/er	47 (7.58)	11 (7.14)	24 (7.52)	12 (8.16)				
Smoker (Yes) (%)	71 (11.40)	18 (11.46)	37 (11.56)	16 (10.96)	0.98 ^†^	0.99 ^¥^	0.99 ^¥^	0.99 ^¥^
Waist Circumference (cm)	93.68 ± 12.64	92.90 ± 11.91	93.49 ± 12.54	94.94 ± 13.56	0.31	0.99	0.28	0.24
Hip Circumference (cm)	103.58 ± 9.32	103.56 ± 9.52	103.44 ± 9.17	103.92 ± 9.48	0.89	0.99	0.98	0.99
BMI (Kg/m^2^)	28.24 ± 4.70	28.27 ± 4.51	28.16 ± 4.51	28.39 ± 5.31	0.96	0.99	0.99	0.99
Systolic Blood Pressure (mmHg)	124.91 ± 15.15	124.81 ± 14.72	124.85 ± 15.07	125.16 ± 15.87	0.95	0.99	0.99	0.99
Diastolic Blood Pressure (mmHg)	78.65 ± 7.70	78.49 ± 7.33	78.94 ± 7.92	78.20 ± 7.60	0.55	0.68	0.99	0.48
Diabetes (Yes) (%)	58 (8.92)	13 (7.98)	23 (6.89)	22 (14.38)	0.02 ^†^	0.99 ^¥^	0.21 ^¥^	0.05 ^¥^
Hypertension (Yes) (%)	261 (42.16)	63 (40.38)	134 (42.14)	64 (44.14)	0.80 ^†^	0.99 ^¥^	0.99 ^¥^	0.99 ^¥^
Ulcer (Yes) (%)	38 (6.14)	9 (5.77)	18 (5.66)	11 (7.59)	0.71 ^†^	0.99 ^¥^	0.99 ^¥^	0.99 ^¥^
Heart Attack (Yes) (%)	15 (2.42)	2 (1.28)	7 (2.20)	6 (4.14)	0.29 ^†^	0.99 ^¥^	0.39 ^¥^	0.88 ^¥^
Stroke (Yes) (%)	2 (0.32)	1 (0.64)	1 (0.31)	0 (0.00)	0.74 ^†^	0.99 ^¥^	--	--
NAFLD (Yes) (%)	393 (60.46)	98 (60.12)	199 (59.58)	96 (62.75)	0.80 ^†^	0.99 ^¥^	0.99 ^¥^	0.99 ^¥^
HOMA	1.98 ± 1.91	1.90 ± 1.08	2.00 ± 2.38	2.03 ± 1.38	0.20	0.29	0.99	0.18
Out of Range (>2.5) (%)	142 (22.12)	35 (22.01)	64 (19.34)	43 (28.29)	0.09 ^†^	0.99 ^¥^	0.61 ^¥^	0.11 ^¥^
Glycemia (mg/dL)	99.14 ± 17.61	98.84 ± 18.15	98.40 ± 17.44	101.07 ± 17.38	0.13	0.99	0.13	0.10
Out of Range (60–110) (%)	91 (14.02)	20 (12.35)	41 (12.28)	30 (19.61)	0.07 ^†^	0.99 ^¥^	0.24 ^¥^	0.14 ^¥^
Insulin (mmol/L)	7.77 ± 4.98	7.50 ± 3.30	7.77 ± 5.56	8.03 ± 5.13	0.21	0.16	0.99	0.35
Out of Range (1.9–23) (%)	13 (2.02)	3 (1.89)	6 (1.81)	4 (2.63)	0.82 ^†^	0.99 ^¥^	0.99 ^¥^	0.99 ^¥^
Cholesterol (mg/dL)	194.96 ± 36.37	197.93 ± 38.22	196.86 ± 34.40	187.69 ± 37.79	0.03	0.96	0.02	0.03
Out of Range (>220) (%)	156 (24.04)	45 (27.78)	81 (24.25)	30 (19.61)	0.23 ^†^	0.99 ^¥^	0.26 ^¥^	0.73 ^¥^
Triglycerides (mg/dL)	105.71 ± 67.45	115.42 ± 72.65	103.04 ± 63.38	101.23 ± 69.76	0.11	0.11	0.09	0.99
Out of Range (36–165) (%)	111 (17.10)	34 (20.99)	53 (15.87)	24 (15.69)	0.32 ^†^	0.53 ^¥^	0.67 ^¥^	0.99 ^¥^
HDL (mg %)	50.86 ± 12.63	50.65 ± 13.02	50.82 ± 12.49	51.16 ± 12.60	0.81	0.99	0.77	0.98
Out of Range (40–60) (%)	253 (38.98)	60 (37.04)	131 (39.22)	62 (40.52)	0.81 ^†^	0.99 ^¥^	0.99 ^¥^	0.99 ^¥^
RBC (10^6^/µL)	4.96 ± 0.52	4.92 ± 0.54	4.98 ± 0.52	4.96 ± 0.51	0.41	0.30	0.45	0.99
Out of Range (3.83–5.08) (%)	244 (37.60)	57 (35.19)	129 (38.62)	58 (37.91)	0.76 ^†^	0.99 ^¥^	0.99 ^¥^	0.99 ^¥^
Hemoblogin (g/dL)	14.04 ± 1.45	13.96 ± 1.49	14.12 ± 1.40	13.95 ± 1.52	0.44	0.43	0.99	0.46
Out of Range (11.7–15.5) (%)	122 (18.80)	33 (20.37)	62 (18.56)	27 (17.65)	0.82 ^†^	0.99 ^¥^	0.99 ^¥^	0.99 ^¥^
Hematocrit (L/L)	42.30 ± 3.72	42.11 ± 3.83	42.46 ± 3.55	42.16 ± 3.95	0.55	0.49	0.99	0.66
Out of Range (34.5–46.3) (%)	94 (14.48)	25 (15.43)	47 (14.07)	22 (14.38)	0.92 ^†^	0.99 ^¥^	0.99 ^¥^	0.99 ^¥^
MCV (fL)	85.77 ± 7.43	86.09 ± 7.24	85.77 ± 7.04	85.44 ± 8.43	0.62	0.52	0.99	0.89
Out of Range (80.4–95.9) (%)	80 (12.33)	20 (12.35)	36 (10.78)	24 (15.69)	0.31 ^†^	0.99 ^¥^	0.99 ^¥^	0.45 ^¥^
MCH (pg)	28.46 ± 2.77	28.54 ± 2.77	28.51 ± 2.62	28.25 ± 3.06	0.65	0.97	0.53	0.80
Out of Range (27.2–33.5) (%)	112 (17.26)	32 (19.75)	51 (15.27)	29 (18.95)	0.38 ^†^	0.68 ^¥^	0.99 ^¥^	0.97 ^¥^
MCHC (g/dL)	33.16 ± 1.08	33.12 ± 1.10	33.23 ± 1.08	33.04 ± 1.03	0.11	0.39	0.65	0.07
Out of Range (32.5–35.2) (%)	176 (27.12)	45 (27.78)	86 (25.75)	45 (29.41)	0.68 ^†^	0.99 ^¥^	0.99 ^¥^	0.99 ^¥^
RDW-CV (%)	13.87 ± 1.48	13.93 ± 1.66	13.80 ± 1.39	13.95 ± 1.51	0.25	0.43	0.99	0.18
Out of Range (10–15) (%)	76 (11.71)	19 (11.73)	37 (11.08)	20 (13.07)	0.82 ^†^	0.99 ^¥^	0.99 ^¥^	0.99 ^¥^
Platelets (10^3^/µL)	230.88 ± 56.01	236.08 ± 65.02	229.02 ± 52.32	229.41 ± 53.50	0.58	0.44	0.78	0.99
Out of Range (159–388) (%)	52 (8.01)	12 (7.41)	24 (7.19)	16 (10.46)	0.44 ^†^	0.99 ^¥^	0.99 ^¥^	0.75 ^¥^
WBC (10^3^/µL)	5.89 ± 1.86	5.73 ± 1.56	5.97 ± 2.11	5.87 ± 1.56	0.60	0.69	0.49	0.99
Out of Range (4.1–11.2) (%)	75 (11.56)	21 (12.96)	42 (12.57)	12 (7.84)	0.26 ^†^	0.99 ^¥^	0.40 ^¥^	0.28 ^¥^
Neutrophils (%)	57.77 ± 7.74	58.28 ± 7.43	57.40 ± 7.93	58.03 ± 7.65	0.50	0.39	0.99 ^†^	0.75 ^†^
Out of Range (39.9–73) (%)	27 (4.16)	7 (4.32)	14 (4.19)	6 (3.92)	0.98 ^†^	0.99 ^¥^	0.99 ^¥^	0.99 ^¥^
Lymphocytes (%)	31.47 ± 7.26	30.95 ± 7.25	31.86 ± 7.37	31.19 ± 7.02	0.55	0.49	0.99	0.64
Out of Range (18.8–50.8) (%)	35 (5.39)	11 (6.79)	15 (4.49)	9 (5.88)	0.54 ^†^	0.94 ^¥^	0.99 ^¥^	0.99 ^¥^
Monocytes (%)	7.36 ± 1.67	7.48 ± 1.73	7.33 ± 1.52	7.30 ± 1.91	0.30	0.60	0.18	0.50
Out of Range (4.1–12.2) (%)	13 (2.00)	4 (2.47)	2 (0.60)	7 (4.58)	0.009 ^†^	0.44 ^¥^	0.94 ^¥^	0.07 ^¥^
Basophils (%)	0.54 ± 0.33	0.53 ± 0.31	0.55 ± 0.35	0.51 ± 0.30	0.20	0.48	0.77	0.13
Out of Range (0.3–1.8) (%)	89 (13.71)	24 (14.81)	40 (11.98)	25 (16.34)	0.38 ^†^	0.99 ^¥^	0.99 ^¥^	0.63 ^¥^
Neutrophils (10^9^/L)	3.45 ± 1.43	3.38 ± 1.16	3.49 ± 1.65	3.43 ± 1.17	0.73	0.99	0.72	0.73
Out of Range (1.8–6.4) (%)	40 (6.16)	8 (4.94)	25 (7.49)	7 (4.58)	0.35 ^†^	0.76 ^¥^	0.99 ^¥^	0.57 ^¥^
Lymphocytes (10^3^/µL)	1.81 ± 0.58	1.73 ± 0.50	1.85 ± 0.60	1.81 ± 0.60	0.23	0.13	0.58	0.75
Out of Range (1.2–3.6) (%)	61 (9.40)	19 (11.73)	27 (8.08)	15 (9.80)	0.42 ^†^	0.64 ^¥^	0.99 ^¥^	0.99 ^¥^
Monocytes (10^3^/µL)	0.43 ± 0.15	0.43 ± 0.17	0.43 ± 0.15	0.42 ± 0.13	0.86	0.98	0.99	0.99
Out of Range (0.3–0.9) (%)	117 (18.03)	35 (21.60)	56 (16.77)	26 (16.99)	0.39 ^†^	0.62 ^¥^	0.90 ^¥^	0.99 ^¥^
Basophils (10^3^/µL)	0.03 ± 0.05	0.03 ± 0.02	0.03 ± 0.07	0.03 ± 0.02	0.42	0.37	0.99	0.51
Out of Range (0–0.2) (%)	1 (0.15)	0 (0.00)	1 (0.30)	0 (0.00)	0.99 ^†^	--	--	--
Eosinophils (10^3^/µL)	0.17 ± 0.15	0.16 ± 0.16	0.17 ± 0.13	0.17 ± 0.17	0.37	0.26	0.40	0.99
Out of Range (0.1–0.5) (%)	202 (31.12)	57 (35.19)	103 (30.84)	42 (27.45)	0.33 ^†^	0.99 ^¥^	0.41 ^¥^	0.99 ^¥^
HbA1c (mmol/mol)	38.08 ± 7.68	38.67 ± 8.67	37.57 ± 6.84	38.55 ± 8.24	0.45	0.61	0.99	0.36
Out of Range (20–42) (%)	113 (17.44)	26 (16.05)	52 (15.62)	35 (22.88)	0.13 ^†^	0.99 ^¥^	0.38 ^¥^	0.20 ^¥^
Total Bilirubin (mg/dL)	0.70 ± 0.36	0.67 ± 0.30	0.70 ± 0.36	0.75 ± 0.41	0.44	0.83	0.31	0.57
Out of Range (0.1–1.2) (%)	65 (10.02)	11 (6.79)	32 (9.58)	22 (14.38)	0.07 ^†^	0.82 ^¥^	0.09 ^¥^	0.42 ^¥^
Direct Bilirubin (mg/dL)	0.17 ± 0.05	0.17 ± 0.05	0.17 ± 0.05	0.18 ± 0.06	0.51	0.99	0.43	0.49
Out of Range (0.1–0.5) (%)	4 (0.62)	1 (0.63)	2 (0.60)	1 (0.65)	0.99 ^†^	0.99 ^¥^	0.99 ^¥^	0.99 ^¥^
GOT (U/L)	22.30 ± 8.67	23.51 ± 14.08	22.23 ± 6.19	21.17 ± 4.87	0.18	0.87	0.11	0.20
Out of Range (0–30) (%)	38 (5.86)	10 (6.17)	22 (6.59)	6 (3.92)	0.50 ^†^	0.99 ^¥^	0.99 ^¥^	0.60 ^¥^
SGPT (U/L)	22.22 ± 12.05	22.28 ± 15.04	22.88 ± 11.89	20.71 ± 8.14	0.17 ^†^	0.38	0.84	0.11
Out of Range (0–30) (%)	77 (11.86)	18 (11.11)	49 (14.67)	10 (6.54)	0.03 ^†^	0.77 ^¥^	0.45 ^¥^	0.01 ^¥^
GGT (U/L)	17.82 ± 13.84	18.72 ± 17.98	17.68 ± 12.77	17.17 ± 10.75	0.61	0.99	0.61	0.52
Out of Range (7–22) (%)	111 (17.10)	26 (16.05)	60 (17.96)	25 (16.34)	0.93 ^†^	0.99 ^¥^	0.99 ^¥^	0.99 ^¥^
Alkaline Phosphatase (U/L)	56.44 ± 16.42	57.31 ± 15.69	56.03 ± 16.58	56.42 ± 16.88	0.34	0.21	0.54	0.99
Out of Range (38–126) (%)	52 (8.04)	13 (8.02)	31 (9.28)	8 (5.30)	0.33 ^†^	0.99 ^¥^	0.99 ^¥^	0.30 ^¥^
Albumin (g/dL)	4.07 ± 0.27	4.08 ± 0.31	4.08 ± 0.26	4.05 ± 0.24	0.72	0.99	0.99	0.64
Out of Range (3.5–5.5) (%)	7 (1.13)	4 (2.58)	1 (0.31)	2 (1.36)	0.06 ^†^	0.25 ^¥^	0.99 ^¥^	0.89 ^¥^
Iron (µg/dL)	90.35 ± 31.20	90.69 ± 32.25	89.46 ± 30.29	91.95 ± 32.15	0.59	0.99	0.70	0.47
Out of Range (49–151) (%)	71 (10.97)	21 (13.04)	34 (10.18)	16 (10.53)	0.62 ^†^	0.99 ^¥^	0.99 ^¥^	0.99 ^¥^
Ferritin (ng/mL)	91.48 ± 83.35	88.09 ± 85.05	95.84 ± 88.09	85.55 ± 69.75	0.34	0.26	0.99	0.48
Out of Range (11–200) (%)	542 (83.51)	134 (82.72)	284 (85.03)	124 (81.05)	0.52 ^†^	0.99 ^¥^	0.99 ^¥^	0.85 ^¥^
Ceruloplasmin (mg/dL)	30.83 ± 6.49	31.83 ± 6.24	30.49 ± 6.43	30.49 ± 6.80	0.04	0.02	0.06	0.99
Out of Range (25–63) (%)	112 (17.26)	16 (9.88)	68 (20.36)	28 (18.30)	0.01 ^†^	0.004 ^¥^	0.09 ^¥^	0.99 ^¥^
α1AT (mg/dL)	172.25 ± 27.36	175.10 ± 28.26	171.00 ± 27.82	171.95 ± 25.26	0.08	0.04	0.17	0.99
Out of Range (90–200) (%)	86 (13.25)	30 (18.52)	40 (11.98)	16 (10.46)	0.07 ^†^	0.19 ^¥^	0.12 ^¥^	0.99 ^¥^

* As Mean and Standard Deviation (M ± SD) for continuous, and as frequency and percentage (%) for categorical variables. ^^^ Kruskal-Wallis rank test or ^†^ Chi-Square or Fisher test; ^Ψ^ Dunn’s test or ^¥^ proportion test for multiple comparisons with Bonferroni adjustment. Abbreviations: BMI, Body Mass Index; NAFLD, Non-alcoholic fatty liver disease; HOMA, Homeostatic Model Assessment; HDL, High-Density Lipoprotein; RBC, Red Blood Cell; MCV, Mean Corpuscular Volume; MCH, Mean Corpuscular Hemoglobin; MCHC, Mean Corpuscular Hemoglobin Concentration; RDW-CV, Red Cell Distribution Width-Coefficient of Variation; WBC, White Blood Cells; HbA1c, Hemoglobin A1c, GOT, Aspartate Amino Transferase; SGPT, Serum Glutamic Pyruvic Transaminase; GGT, Gamma-Glutamyl Transferase; α1AT, Alpha-1-Antitrypsis.

**Table 2 ijms-25-09512-t002:** Logistic regression model ^ of diabetes on rs2802292.

Genotypes	OR	se (OR)	95% C.I.	*p*
GG [*Ref.*]	--	--	--	--
GT	0.95	0.35	0.46 to 1.98	0.90
TT	2.14	0.82	1.01 to 4.53	0.05
*G-Carriers*				
TT [*Ref.*]	--	--	--	--
GG/GT	0.45	0.13	0.25 to 0.81	0.008

Abbreviations: OR, Odds Ratio, se (OR), Standard Error of OR; 95% C.I. Confidence Intervals at 95%. ^ Models adjusted for age and gender.

**Table 3 ijms-25-09512-t003:** Mediation analysis ^ of rs2802292 on NAFLD and diabetes as mediator.

Effects	β	se (β)	95% C.I.	*p*
*Indirect*				
rs2802292 (GT) → Diabetes → NAFLD	−0.001	0.006	−0.01 to 0.01	0.84
rs2802292 (TT) → Diabetes → NAFLD	0.01	0.008	−0.001 to 0.03	0.08
*Component*				
rs2802292 (GT) → Diabetes	−0.01	0.03	−0.06 to 0.05	0.83
rs2802292 (TT) → Diabetes	0.10	0.03	0.003 to 0.12	0.04
Diabetes → NAFLD	0.13	0.06	0.10 to 0.35	<0.001
*Direct*				
rs2802292 (GT) → NAFLD	0.008	0.04	−0.08 to 0.09	0.85
rs2802292 (TT) → NAFLD	0.009	0.05	−0.09 to 0.11	0.84
*Total*				
rs2802292 (GT) → NAFLD	0.007	0.04	−0.08 to 0.09	0.88
rs2802292 (TT) → NAFLD	0.02	0.05	−0.08 to 0.13	0.63

^^^ Model adjusted for age and gender.

## Data Availability

The original contributions presented in the study are included in the article. Further inquiries can be directed to the corresponding author.

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
