# Peer review of "Exploring the Relationship of rs2802292 with Diabetes and NAFLD in a Southern Italian Cohort—Nutrihep Study"

_ijms, 2024, doi:10.3390/ijms25179512_

Round 1

Reviewer 1 Report

Comments and Suggestions for Authors

The authors presented the relationship of rs2802292 with diabetes and NAFLD in a Southern Italian cohort. This FOXO3 has been found to increase lifespan and regulate insulin pathways. The authors recently found a remarkably higher cancer risk for Peutz-Jeghers and the PTEN hamartoma tumor syndrome in TT genotype carriers of this rs2802292 allele. They concluded FOXO3 variant have protective effects against diabetes and NAFLD. Furthermore, they found that no direct relationship between rs2802292 and NAFLD, but the effect is direct on diabetes. These results might be a novel and useful information if these had been correctly analyzed and interpreted in this manuscript. Yu Qin Mao already reported that none of the 4 FOXO3 variants, rs13217795, rs2764264, rs2802292, and rs13220810, were associated with the risk of type 2 diabetes compared to controls (Med Sci Monit 2019 22:25:2966-2975). A total of 843 elderly residents from east China were enrolled in this study, which included 426 patients with type 2 diabetes and 417 controls.  Unfortunately, Forte G’s manuscript included only 57 patients with diabetes and there were no differences between GG allele and TT allele about HbA1c and glucose. This is a fatal point, and the authors should be more cautious in their interpretations of these results. It is not enough to gain novel conclusion by a small number of patients even if the significant differences were provided.

Author Response

Dear Editor,

we are pleased to submit the amended version of our work “Exploring the relationship of rs2802292 with diabetes and NAFLD in a Southern Italian cohort. Nutrihep study” (ijms-3144359), which we would like to have considered for publication in International Journal of Molecular Sciences as part of the special issue “Recent Advances in the Diagnosis and Treatment of Chronic Liver Diseases 2023”. We addressed below all the comments raised by the Reviewers, mainly by responding to their observations.

Reviewer reports:

Reviewer #1: The authors presented the relationship of rs2802292 with diabetes and NAFLD in a Southern Italian cohort. This FOXO3 has been found to increase lifespan and regulate insulin pathways. The authors recently found a remarkably higher cancer risk for Peutz-Jeghers and the PTEN hamartoma tumor syndrome in TT genotype carriers of this rs2802292 allele. They concluded FOXO3 variant have protective effects against diabetes and NAFLD. Furthermore, they found that no direct relationship between rs2802292 and NAFLD, but the effect is direct on diabetes. These results might be a novel and useful information if these had been correctly analyzed and interpreted in this manuscript. Yu Qin Mao already reported that none of the 4 FOXO3 variants, rs13217795, rs2764264, rs2802292, and rs13220810, were associated with the risk of type 2 diabetes compared to controls (Med Sci Monit 2019 22:25:2966-2975). A total of 843 elderly residents from east China were enrolled in this study, which included 426 patients with type 2 diabetes and 417 controls.  Unfortunately, Forte G’s manuscript included only 57 patients with diabetes and there were no differences between GG allele and TT allele about HbA1c and glucose. This is a fatal point, and the authors should be more cautious in their interpretations of these results. It is not enough to gain novel conclusion by a small number of patients even if the significant differences were provided.

We thank the Reviewer for these comments and remarks about the paper by Mao et al. (cited in our manuscript, see ref.43). In this paper, Mao et al. showed that the longevity-associated FOXO3 variants were correlated with lower blood glucose levels in elderly women with type 2 diabetes in east China.  Unfortunately the paper is not very comparable with our study. As Mao et al. cited “A total of 843 elderly residents from east China were enrolled in this study. There were 426 T2D patients and 417 control participants.” The paper suggested, different to our study based on observational evaluation in a defined observational period (2005-2006), considers a Chinese cohort of a specific geographical area reporting a prevalence of diabetes of 50.53% (or an absolute frequency of 426 subjects on 843), which is the opposite of other epidemiological studies conducted in China. This prevalence is not representative of the Chinese population, but only of the cohort considered by the authors, moreover in 2 strangely balanced arms, without referring to a possible initial sample size. Official sources report a prevalence of 13% of diabetes in China in 2021 [https://idf.org/our-network/regions-and-members/western-pacific/members/china/], while studies in the literature show an increasing trend from 1980 to 2013 that reaches a prevalence of 10.90% [Hu C., Jia W. Diabetes in China: Epidemiology and Genetic Risk Factors and Their Clinical Utility in Personalized Medication. Diabetes. 2018;67(1):3-11]. While more recent data report a projected for the years 2020-2030 in Chinese adults (aged 20–79 years) from 8.2% to 9.7% [Liu J., Liu M., Chai Z., et al. Projected rapid growth in diabetes disease burden and economic burden in China: a spatio-temporal study from 2020 to 2030. Lancet Reg Health West Pac. 2023:33:100700]. Our total cohort report a prevalence of diabetes of 9.21%, i.e. 57 participants (in epidemiology is often preferred using prevalence, i.e  the proportion of individuals in a population or cohort, when comparing the burden of disease because it normalizes the data relative to total cohort size, and not absolute frequency that refers to the actual number of cases or events in a cohort without considering the size of the population or cohort), in line with worldwide data [Sun H., Saeedi P., Kararunga S., et al. IDF Diabetes Atlas: Global, regional and country-level diabetes prevalence estimates for 2021 and projections for 2045. Diabetes Res Clin Pract. 2002:183:109119].

As with most epidemiological studies, these must focus on a particular population that lives in a specific geographical area. This type of approach is crucial for understanding health outcomes and disease patterns within a defined population. In addition, our paper aims to demonstrate how NAFLD is a “storage disease”, that is, a disease that is the result of more immediate imbalances that occur over time (i.e. diabetes). In fact, to emphasize this, a mediation analysis was performed. Furthermore, it is well known and widely described that the rs2802292 has stronger effect on longevity in Asian and European [Sun L., Hu C., Zheng C., Qian Y., Liang Q., Lv Z., Huang Z., Qi K., Gong H., Zhang Z., Huang J., Zhou Q., Yang Z. FOXO3 variants are beneficial for longevity in Southern Chinese living in the Red River Basin: a case control study and meta-analysis. Sci Rep 2015; 5:9852] and is implicated in protection against chronic conditions of aging (i.e. diabetes, cardiovascular disease, stroke). Furthermore this is the first time that a molecular analysis were conducted on this cohort.

Reviewer #2: Certainly an interesting study, bringing another of the parameters that determine or could determine longevity.

The introduction is apt. It mentions the involvement of the FOXO3 gene hat FOXO3 regulates the expression of genes controlling cell cycle gene expression, DNA damage repair, energy metabolism, oxidative stress resistance, apoptosis which is a lot and very complex events.

The materials and methods section points to a systematic long-term study. In addition to genetic analysis, anthropometric measurements and biochemical parameters were performed as part of routine screening examinations. Considering the description in the introduction, their mentioned previous studies, the authors could also expand into investigations of other, not common parameters.

The description of the results, statistical evaluation and discussion is adequate to the findings. I am even more inclined to believe that the finding that  a direct effect of rs2802292 SNP with NAFLD only in TT genotypes can be even more supported by examination of SOD along with ceruloplazmin.

However, it is not a lack of paper as a whole.

We thank the Reviewer for their thoughtful feedback on our article and appreciate their suggestion about the potential future direction of our research, in particular the idea of implementing the investigations of other, not common parameters. This suggestion aligns well with our goal of advancing our findings toward clinical practice, and we will consider integrating this approach into our future studies.

Please see the attachment (pdf copy)

Reviewer 2 Report

Comments and Suggestions for Authors

Certainly an interesting study, bringing another of the parameters that determine or could determine longevity.

The introduction is apt. It mentions the involvement of the FOXO3 gene hat FOXO3 regulates the expression of genes controlling cell cycle gene expression, DNA damage repair, energy metabolism, oxidative stress resistance, apoptosis which is a lot and very complex events.

The materials and methods section points to a systematic long-term study. In addition to genetic analysis, anthropometric measurements and biochemical parameters were performed as part of routine screening examinations. Considering the description in the introduction, their mentioned previous studies, the authors could also expand into investigations of other, not common parameters.

The description of the results, statistical evaluation and discussion is adequate to the findings. I am even more inclined to believe that the finding that  a direct effect of rs2802292 SNP with NAFLD only in TT genotypes can be even more supported by examination of SOD along with ceruloplazmin.

However, it is not a lack of paper as a whole.

Author Response

(The authors gave the same response as above.)

Round 2

Reviewer 1 Report

Comments and Suggestions for Authors

I understand what the authors want to say and how they got conclusion. 

Author Response

We thank the Reviewer for his/her thoughtful feedback on our article